# Early bilingual immersion school program and cognitive development in French-speaking children: Effect of the second language learned (English vs. Dutch) and exposition duration (2 vs. 5 years)

**Sophie Gillet * , Cristina Barbu, Martine Poncelet**

Psychology and Neuroscience of Cognition Research Unit, University of Liège, Quartier Agora (B33), Liège, Belgium

* s.gillet@uliege.be

## Abstract

The results of studies targeting cognitive and academic advantages in children frequenting early bilingual immersion school programs (CLIL) have been contradictory. While the impact of the amount of CLIL experience has already been studied, the role of the second language learned has been little studied to account for differences among study findings. The link between executive skills (EF) and scholar abilities (e.g., mathematics) in the CLIL context has also been little investigated. The purpose of the present study was to determine if the impact of CLIL on EF and academic performances varies depending on the immersion language and the duration of CLIL experience. The sample included a total of 230 French-speaking children attending second (141) and fifth (89) grade classes. Within each grade, there were three matched language groups composed of children respectively immersed in English, immersed in Dutch, and non-immersed controls. The children were administered tasks assessing executive functions [alerting, cognitive flexibility, and working memory], as well as arithmetic abilities. In second grade, we detected no difference in EF between the language groups. On the other hand, in fifth grade, the two immersed groups outperformed the non-immersed group on the cognitive flexibility task but did not differ between them. Moreover, only the Dutch immersed group outperformed the control group on the working memory task. Arithmetic performances also differed depending on the language learned; in second grade, Dutch learners performed better than the monolingual group. In fifth grade, Dutch learners outperformed the two other groups. These results suggest that the impact of CLIL on executive skills and arithmetic performances might be modulated by the amount of CLIL experience and the second language learned in immersion.

## Introduction

Bilingual immersion school programs provide an environment of intensive exposure to the second language (L2) and opportunities to use the L2 in ecologically authentic contexts. All

**Data Availability Statement:** Data is available in the following project that you will find on my

ORCID profile: 0000-0002-1688-319X on https://osf.io/unbmf/.

**Funding:** The author(s) received no specific funding for this work.

**Competing interests:** The authors have declared that no competing interests exist.

children starting at the same age and receiving the same amount of input in the same context (school) makes Content and Language Integrated Learning (CLIL) approaches of particular interest to evaluate the L2 learning effect on executive functions in a homogeneous population and context. In recent years, cognitive advantages have been reported for children attending a bilingual immersion program [for e.g., 1–4]. Recently, some researchers [5] evaluated French-speaking children learning Dutch at the beginning and at the end of CLIL schooling on alerting, selective auditory attention, divided attention, cognitive flexibility, and working memory tasks. These authors found no advantages on any tasks at the beginning of the schooling (first, second, and third grades), but did find advantages in cognitive flexibility and working memory tasks at the end of schooling (sixth grade). Previous studies [2,3,6] using the same tasks, except the working memory task, evaluated French-speaking children learning English and found, contrarily, advantages at the beginning but not at the end of the CLIL schooling. These authors detected an advantage in selective auditory attention in first grade [3], and in alerting, selective auditory attention, divided attention, and cognitive flexibility in third grade [2], but no advantage at the end of CLIL schooling in sixth grade [6]. Thus, the results of studies using the same tasks conducted with French-speaking children learning English or Dutch as L2 show different cognitive advantages emerging at different moments of the schooling.

Among the other studies evaluating the impact of CLIL context on cognitive development inconsistent results were found [4,7–9]. In these studies different pairs of languages were tested (e.g., Spanish-English, Serbian-English), but other factors, such as the tasks used, the CLIL methodology, the manner to match the immersed and non-immersed groups, also varied, making it difficult to interpret the role of the languages at stake on executive functions (EF) advantages. Moreover, to our knowledge, there is no published study directly comparing the impact of the languages learned to determine the influence of the language pair on the EF performance in the CLIL context. Yet, the learning of different languages could solicit EF differently at different moments of the CLIL. In fact, previous studies evaluating French-speaking children learning English vs. Dutch suggested that the second language learned could lead to different outcomes. These two languages present some differences both at the linguistic and non-linguistic levels.

In terms of linguistic characteristics, the English-French pair is more similar than the Dutch-French pair. Although English and Dutch are two Germanic languages, they present differences, especially at the lexical and syntactic levels. At the lexical level, English is closer to French than to Dutch, given their shared history, which has led to reciprocal lexical loans. For example, the French-English pair (5206) has double the amount of translation equivalents frequency as the French-Dutch pair (2599) [10]. The degree of relatedness between languages seems to be strongly associated with the frequency of cognates, and this frequency is much higher in the French-English pair than in the French-Dutch pair. This similarity could induce more cognitive control in word selection in French-English bilinguals.

At the sentence level, the underlying syntactic structure in Dutch is different from those of French and English, which are more similar to each other. In Dutch, some verbal forms (e.g., infinitives and past participles) are placed at the end of the sentence. This is also the case in sub-clauses in which all verbal forms are rejected at the end of the sentence. Thus, Dutch is said to be a head-final (Subject-Object-Verb) language, whereas English or French are head-initial (Subject-Verb-Object) languages [11]. These syntactic differences should induce facilitation in L2 processing in the case of similar syntactic structures (French-English) and/or modulate some cognitive functions, especially working memory in the case of head-final structure (Dutch) when L1 is head-initial (French).

In most Germanic languages (like Dutch, but far less English), verb-second word order (V2) is found, which means that regardless of whether the main clause starts with a subject (S)

or with another fronted element (X), such as an adverbial, the finite verb (V) appears in the second position. This is referred to as subject-verb inversion or XVS word order [12]. Using the ERP technique (P600-effect), Andersson et al. [12] have examined how adult learners of Swedish process online this V2 in the L2 depending on whether their L1 has V2 (German) or not (English) relative to Swedish native speakers. These authors found that the presence of similar word order in the native language and the second language learned facilitated the processing of the L2. Thus, cognitive pressure of being bilingual could differ depending on the proximity between the mother tongue language (L1) and the second language learned (L2) in facilitating or not the processing of the L2. This easier processing could induce a faster L2 mastering and lead to earlier cognitive advantages in children learning English as L2. Furthermore, the syntactic structure of a language has been suggested to influence the manipulation abilities in working memory (WM) by maintaining either the beginning or the end of a sentence to catch the head of it (verb). In this regard, Amici et al. [13] showed that specific characteristics (i.e., the syntax and word order) of our native language might predict the way we process, store, and retrieve information. We might expect that the characteristics of the L2 intensively learned should also have an impact on working memory, such that French-speaking children learning Dutch (but not English) could enhance their manipulation abilities in working memory along with L2 practice.

Besides these linguistic aspects, another difference between Dutch and English is that English is far more present in the environment of most children (e.g., video games, music, social media) than Dutch. And this extra scholar exposition has, for example, been shown to lead 11-year-old Dutch-speaking children to acquire English knowledge before any formal scholar lessons in English [14]. This could also increase the switching opportunities of the child between his two languages. Moreover, this switching behaviour has been linked with the advantage found in cognitive flexibility tasks [15,16]. The children learning English as L2 could thus present the bilingual cognitive advantages earlier in their schooling.

In the languages examined in the present study, the English-French pair is more similar than the Dutch-French pair at multiple levels. These differences (or similarities) could induce differences in the EF outcomes. Learning English or Dutch for French-speaking children could have a different influence on cognition in that they differ in their linguistic characteristics (i.e., lexical similarity, orthography transparency and word order) but also in their extra scholar exposition, as well as in the opportunities to use them ([5,14]).

Concerning mathematics, many studies suggest that higher EF performances are linked with higher academic performances [e.g., 17–21]. The studies that compare the arithmetic abilities between immersed and non-immersed children showed mostly an advantage for immersed children although some mixed results were found [22,23]. Marian, Shook, and Schroeder [24] have, for example, reported a superiority in arithmetic tasks in English or Spanish speaking children frequenting a CLIL program in Spanish and English, respectively, in comparison to non-immersed controls in grades 3, 4, and 5. Fleckenstein, Gebauer, and Möller [25] also showed that arithmetic abilities of German-speaking children immersed in English increased greater and faster than those of non-immersed children.

Based on these results, we could hypothesize that immersion could have a positive impact on arithmetic performances.

Moreover, the comparison of the performance in the arithmetic of children learning English or Dutch could also be of particular interest. As for the syntactic level in the sentence processing, Dutch vs. French or English number naming structures differ. For example, a two-digit number name follows a unit-ten order in Dutch (drieentwintig = three and twenty for twenty-three) but a ten-unit order in French (vingt-trois = twenty-three) like in English [26]. This language-specific number word structure could also induce modulations in a bilingual's

EF and/or arithmetic performances. Children have, in the case of learning mathematics in Dutch, which has the reverse structure of French structure (21 is said twenty-one in English or vingt-et-un in French, but said eenentwintig in Dutch = one and twenty), to manipulate two different syntactic structures of the number. This early verbal manipulation (from kindergarten that corresponds to the beginning of the CLIL class) could, in turn, induce a better and faster concept formation of units and tens. This could have a positive influence on further calculation abilities. Moreover, the studies evaluating the influence of linguistic properties on processing place-value information showed that number word inversion leads to additional processing costs in various numerical tasks (e.g., multi-digit addition) [27]. This early additional processing could further train cognitive abilities like working memory that is known to predict arithmetic abilities [e.g., 28].

Whereas Barbu, Gonzalez, Gillet, & Poncelet [29] showed that French-speaking children immersed in English since two years did not present an advantage on executive functions nor on addition calculations, they showed, surprisingly, a disadvantage for immersed children in subtraction calculations. According to these authors, as the immersed children do not master their L2 sufficiently, they use important attentional resources when processing L2 information given by teachers. As a consequence, the constant L2 processing might have generated an extra-cognitive load during the task that might have a negative impact on the completion of arithmetic operations [29]. Further investigations should permit better understanding of arithmetic's evolution in immersed children.

The aim of this study was to determine the impact of the second language learned and the time spent in the CLIL program [30] on cognitive and arithmetic development. Therefore, we compared alerting, cognitive flexibility, working memory and arithmetic performances of three language groups of children immersed in English, in Dutch (since the third kindergarten) and non-immersed controls, respectively, at the beginning (grade 2) and at the end (grade 5) of the elementary school. Within each grade, the three language groups were matched on age, gender, SES, verbal, and nonverbal abilities.

Based on previous research evaluating French-speaking children, our expectations are detailed in the four paragraphs below.

Concerning alerting, the study aims to provide more information as to whether alerting is affected by immersion or not. An advantage could be observed in the first stages of CLIL because these skills could be solicited as children are in a continuous readiness state to process an L2 and on the search of (visual) cues that could help them to understand the discourse. As advantages were found in English [1,2] in third grade but not in sixth grade [6] and no advantage was found in Dutch in sixth grade [5], we would determine if an advantage could be found in fifth grade in one (or both) of the two language groups with increased CLIL experience.

Concerning cognitive flexibility, an advantage on this function was expected as the children are frequently in situations in which they are required to switch from one language to another in the CLIL context. We hypothesized that children immersed in English could show an earlier EF advantage [1,2] than those immersed in Dutch [5,8] in comparison with non-immersed children, perhaps because of the higher similarity in this pair of languages (French-English). In previous studies, in English immersed children, an advantage was found from third grade [1–3,29] but not in sixth grade [6]. This illustrates the fluctuations of the cognitive advantages during CLIL schooling. The present study could address whether an advantage could still be present in fifth grade. In a previous study, in Dutch immersed children, an advantage was found in sixth grade but not before (first, second and third grade) [5]. Therefore, it would be important to determine when during the primary schooling, between third grade and sixth grade, advantages in cognitive flexibility emerge. The present study evaluating AEF at the

beginning and at the end of schooling allow us to determine if advantages are already present in fifth grade or not.

Concerning working memory, an advantage was expected as the immersed children must maintain the (L2) information for longer to make sure to have the time to translate, infer, and use the context cues, to understand the discourse in CLIL context. We hypothesized an advantage for immersed children whatever the L2 learned as previous studies showed advantages at the beginning [3,4] and at the end [5] of CLIL schooling in different language pairs. However, the differences in terms of word order in French and in Dutch sentences could train working memory abilities more in children learning Dutch than English. In Dutch immersed children, an advantage was found in sixth grade but not in first, second, and third grade [5]. Therefore, it is important to determine at which point during CLIL schooling such benefits emerge and consequently if this advantage could already be present 1 year before, in fifth grade. In English immersed children, no study (to our knowledge) has evaluated working memory in French-speaking children. We therefore would evaluate this pair of languages to determine whether the second language learned could modulate the advantage in working memory.

As in Dutch the inverse naming structure of the number could affect children's representation of the ten-unit structure of numbers, we should expect an additional advantage for children immersed in this L2 particularly. As a reminder, with English immersed children, Barbu et al. [29] showed a disadvantage in second grade despite an absence of difference on EF in comparison with non-immersed children. This study intends to determine whether a disadvantage really exists or not in this pair of languages and, if so, whether if this disadvantage is also present in fifth grade.

## Method

### Participants

A sample of 141 typically developing French-speaking children in Grade 2, composed of 47 children following a Dutch school program (Immersed in Dutch; ImD), 46 children following an English school program (Immersed in English; ImE), and 48 non-immersed monolinguals (NonIm), was tested. Another sample of 89 typically developing French-speaking children in Grade 5 was composed of 30 children following a Dutch immersion program, 29 children attending an English immersion program, and 30 non-immersed monolinguals participated in the study. All the immersed children were following a school immersion program since their third kindergarten (5 years old) with an exposition varying between 50 and 75% of the curriculum in L2. Each group was composed by children that came from four to five different schools. The children were all tested in the morning, individually in a quiet room, in their school. There were three experimenters for the children in fifth grade, and four experimenters for the children in second grade. The experimenters had to test the same number of children from each language group in order to avoid an experimenter effect. French was the language of testing used for all the children. The different language groups of each grade (second and fifth) were matched on age, socio-economical level (SES), gender, and nonverbal and verbal intelligence abilities. The participants were recruited from traditional and immersion schools in the French-speaking community of Belgium. The sample characteristics for each grade are presented in Tables 1 and 2. On the basis of a parental questionnaire, we exclusively included children that were native speakers of French, did not suffer from neurological disorders or sensory deficits, and did not present a history of speech or language impairment, and we excluded children speaking two languages at home or following extra-scholar lessons in a second language. From the parental questionnaire, we also got the socioeconomic status (SES) of the family. Concerning SES, we used the highest level of education of the parents as a proxy for

**Table 1. Sample characteristics (Gender and SES), descriptive statistics, and Chi$^2$ for comparison in gender, SES and nonverbal reasoning indicators (N = 230).**

| | Grade 2 (n = 141) | | | Grade 5 (n = 89) | | |
|---|---|---|---|---|---|---|
| | ImD (n = 47) | ImE (n = 46) | NonIm (n = 48) | ImD (n = 29) | ImE (n = 30) | NonIm (n = 30) |
| Gender Ratio (m:f) | 24:23 | 25:21 | 23:25 | 14:15 | 11:19 | 13:17 |
| Test Chi$^2$ for gender | $X^2$ (2) = 3.88, p = .82 | | | $X^2$ (2) = 0.81, p = .66 | | |
| Sociocultural level* | | | | | | |
| 1 | 0 | 0 | 1 | 0 | 0 | 1 |
| 2 | 13 | 15 | 11 | 2 | 9 | 5 |
| 3 | 20 | 16 | 18 | 14 | 13 | 10 |
| 4 | 14 | 15 | 18 | 13 | 8 | 14 |
| Test Chi$^2$ for SES | $X^2$ (6) = 3.54, p = .73 | | | $X^2$ (6) = 9.07, p = .16 | | |
| Nonverbal reasoning Raven | | | | | | |
| 0 (= p5) | 0 | 0 | 0 | 2 | 2 | 2 |
| 1 (p5-p10) | 0 | 1 | 1 | 2 | 1 | 0 |
| 2 (p10-p25) | 3 | 6 | 1 | '7 | 7 | 9 |
| 3 (p25-p50) | 12 | 17 | 18 | 10 | 9 | 9 |
| 4 (p50-p75) | 12 | 12 | 13 | 8 | 6 | 8 |
| 5 (p75-p90) | 15 | 7 | 12 | 0 | 4 | 2 |
| 6 (p90-p95) | 5 | 3 | 3 | 1 | 0 | 0 |
| Test Chi$^2$ for nonverbal reasoning | $X^2$ (10) = 9.78, p = .45 | | | $X^2$ (12) = 8.75, p = .72 | | |

*1 = 6 years and +; 2 = 12 years and +; 3 = high school level; 4 = university level.

socioeconomic status. Our three groups were divided into four categories in the function of the parent that has the higher diploma: 1 = primary; 2 = secondary; 3 = high degree; 4 = university degree. None of the schools (immersion and non-immersion) displayed 'active' pedagogic curricula known to improve executive functioning [31].

The children of the different language groups were further matched on the nonverbal reasoning abilities and the French lexical receptive abilities because they are both measures of children's conceptual development that is linked to attentional control functioning [e.g., respectively 32–34].

## Tasks

**Background measures.** *French lexical receptive abilities*. The French adaptation of the Peabody Picture Vocabulary Test-Revised [35], the *Échelle de vocabulaire en images Peabody* [EVIP; 36], was used to evaluate the participants' French receptive vocabulary. Single words were presented to the child orally in the presence of four drawings. The child was asked to select the one that best matches the word. The standard procedure of notation was followed. We used the standardized mean scores in the analysis.

**Table 2. Sample characteristics (age), descriptive statistics, and t-tests of comparison in age and verbal intelligence indicators.**

| | Grade 2 (n = 141) | | | Grade 5 (n = 89) | | |
|---|---|---|---|---|---|---|
| | ImD Mean (SD) | ImE Mean (SD) | NonIm Mean (SD) | ImD Mean (SD) | ImE Mean (SD) | NonIm Mean (SD) |
| Age (month) | 91.5 (3.69) | 90.9 (3.08) | 91.2 (3.78) | 127.7 (4.34) | 127.7 (3.24) | 128.8 (3.83) |
| | $F(2,138) = 0.33$; p = .71; ES < 0.01 | | | $F(2,86) = 1.44$; p = .24; ES < 0.01 | | |
| EVIP (standardized mean scores) | 116.2 (12.1) | 113.6 (12.0) | 116.7 (12.5) | 113.0 (10.4) | 112.8 (7.8) | 110.8 (15.4) |
| | $F(2,138) = 0.84$; p = .43; ES = 0.01 | | | $F(2,86) = 0.32$; p = .72; ES = 0.03 | | |

*Nonverbal intelligence abilities*. The coloured version of Raven's Progressive Matrices [37] was administered to the participants to assess nonverbal reasoning abilities. The standard procedure of notation was used. We used the percentile scores in the analysis.

**Executive functions measures.**   We used tasks evaluating alerting and cognitive flexibility provided from standardized batteries: a child version for children from 6 to 10 years old [KITAP, 38; French adaptation] and an adult version for older children and adults [TAP, 39]. The children version was used for children in Grade 2, and the adult version (TAP) was used for children in Grade 5. We used a backward digit span evaluating working memory [40, Wechsler Intelligence Scale for Children-Fourth Edition, WISC-IV]. The same task was used in grades 2 and 5. As in previous studies [3,5] using these batteries, we used median reaction time and not mean reaction time because this measure is less affected by a short potential distraction during task administration. Consequently, we assume that median reaction times better reflect the child's actual performance.

*Alerting*. In the alerting task from the KITAP, a witch appeared in the middle of the computer screen. Children were demanded to press a response key as fast as possible when the stimulus (the witch) was appearing. In the alerting task from the TAP, a cross replaces the witch. Correct responses and median reaction times served as dependent variables.

*Cognitive flexibility*. In the cognitive flexibility from the KITAP, two dragons are simultaneously presented on the computer screen, a green dragon and a blue dragon. Children could press on two reaction keys, one with the left hand and the other with the right hand. They were asked to alternatively react to the blue and then to the green dragon by pressing the reaction key that was localized in front of the target dragon. The side on which the target would appear was unpredictable. In the cognitive flexibility from TAP, the dragons are replaced by letters and numbers. The participant had to react to the number and the letter alternatively by pressing the right reaction key (in front of the target). The number of correct responses and median reaction times served as the dependent variable.

*Working memory*. In the backward digit span task (WISC-IV), participants were hearing a digit sequence and were required to repeat it in reverse order. The sequences get progressively longer, ranging from two to eight digits maximum. The number of sequences correctly repeated was used as the dependent variable.

**Arithmetic achievement measures.**   *Arithmetic number fact problems*. In the Tempo Test Rekenen [TTR; 41], which is a paper-and-pencil arithmetic facts test consisting of 200 arithmetic number fact problems (e.g., 13 + 9 = _) divided into five lists (additions, subtractions, multiplications, divisions, and mixed), children had to solve as many number-fact problems as possible out of each list within 1 min. For children in Grade 2, we only administered the additions and subtractions, and for children in Grade 5, the five types of arithmetic number facts were proposed. The number of correct responses for each list was used as the dependent variable. This tool was used because it involves less verbal skills than problem solving, for example. Indeed, arithmetic facts and language functions are thought to be relatively independent systems [e.g., 42–44]. A problem presented in the Arabic format is likely to activate the same non-verbal codes for both French- and Dutch-speaking children, consequently, no language difference is expected [45]. As a consequence, this test will be less sensitive to the level of mastery of L2 and to the language of teaching arithmetic that can vary in the different schools of immersion (English or Dutch in respectively English or Dutch bilingual schools).

**General procedure.**   The children performed different tasks over a set of two sessions (approximately 20 min per session) in a fixed order. The order and distribution per session are described in Table 3. Children were tested individually in a quiet room in their respective schools during the second semester of the school year (from February to April). They all were tested during the morning.

**Table 3. Order of presentation and distribution of the tests by session.**

| Session 1 | Session 2 |
|---|---|
| Alerting [38,39] | Cognitive flexibility [38,39] |
| Working memory task [WISC-IV, 40] | Verbal intelligence [EVIP– 36] |
| Non-verbal intelligence [37] | TTR [41] |

## Ethics statement

Each pupil participated voluntarily, and parental consent was obtained. The study had received approval from the committee on ethics of the Faculty of Psychology, Speech Therapy, and Education Sciences from the University of Liège.

## Results

To investigate the differences in the cognitive abilities according to the language group (ImD, ImE, NonIm), we conducted an analysis of variance (ANOVA) with the groups as an independent factor and the performances on the different cognitive measures as a dependent factor in grades 2 and 5.

We also used Bayesian statistics to control for biases related to the normal distribution of data, the null hypothesis, statistical power, or p-values [46,47]. This approach allows for a comparison of two models (group effect compared to a null model) using the Bayesian factor. This factor reflects the probability of occurrence for these two models. The level of significance of the Bayesian factor is not related to a threshold value as in inferential statistics. It is generally acknowledged to consider a Bayesian factor greater than 3 as moderate evidence, a Bayesian factor over 10 as strong evidence, and a Bayesian factor more than 30 as very strong evidence [48].

### Performance on executive functions tasks in different language groups

The descriptive statistics in terms of median reaction times and correct responses concerning attentional and executive tasks are detailed below. As the tasks used in each grade were slightly different, we presented the results from grade 2 and those of grade 5 separately. Table 4 summarizes the results on the different tasks administered, separately for grades 2 and 5, for the

**Table 4. Means (Standard deviations) of the different tasks administered, by grade.**

| | Grade 2 | | | Grade 5 | | |
|---|---|---|---|---|---|---|
| | **ImD (n = 47)** | **ImE (n = 46)** | **NonIm (n = 48)** | **ImD (n = 29)** | **ImE (n = 30)** | **NonIm (n = 30)** |
| Tasks | KiTAP Mean (SD) | | | TAP Mean (SD) | | |
| Alerting | | | | | | |
| Median time (ms) | 355 (63) | 363 (72) | 366 (72) | 314 (23) | 295 (44) | 313 (39) |
| Cognitive flexibility | | | | | | |
| Correct responses | 42.2 (4.4) | 42.0 (5.2) | 41.9 (4.4) | 85.3 (8.3) | 87.8 (8.1) | 80.6 (10.9) |
| Median time (ms) | 1157 (256) | 1153 (276) | 1109 (257) | 1000 (201) | 957 (332) | 1103 (305) |
| | Same task (repeating digits in inverse order) | | | | | |
| Working memory | | | | | | |
| Number of correct sequences | 6.4 (1.2) | 6.2 (1.2) | 6.0 (1.2) | 7.6 (1.4) | 7.1 (1.8) | 6.4 (1.3) |
| Span | 3.7 | 3.5 | 3.4 | 4.4 | 3.9 | 3.6 |

**Table 5. BF resulting from Bayesian ANOVAs on alerting, cognitive flexibility, and working memory measures in Grade 2.**

| Variable | Model | BF10 | BF01 | error % |
|---|---|---|---|---|
| Alerting (RT) | Group effect (Im or NonIm) | 0.088 | **11.300** | 0.024 |
| Cognitive flexibility (RT) | Group effect (Im or NonIm) | 0.105 | **9.502** | 0.024 |
| Cognitive flexibility (CR) | Group effect (Im or NonIm) | 0.071 | **13.996** | 0.024 |
| Working memory (CR) | Group effect (Im or NonIm) | 0.152 | **6.572** | 0.025 |

*Note*. CR = Correct Response; RT = Reaction Times; $BF_{10}$ = Bayes factor for the alternative hypothesis vs. the null hypothesis; BF 01 = Bayes factor for the null hypothesis vs. the alternative hypothesis.

immersed groups and the non-immersed groups. Tables 5 and 6 present the Bayesian ANO-VAs results in Grade 2 and Grade 5, respectively.

**Alerting. In second grade**, no analyses were realized on correct responses because of a ceiling effect. Concerning median reaction times, no effect of language group was found ($F$ (2.138) = 0.28, $p$ = .75, $\eta p^2$ 0.004). The absence of effect of the group is confirmed by Bayesian statistics ($BF_{10}$ = 0.08).

**In fifth grade**, no significant language group effect was found on median reaction times ($F$ (2.86) = 2.61, $p$ = .07, $\eta p^2$ 005). Newman-Keuls post hoc analysis and Bayesian post hoc comparisons were applied. No significant difference was found between the groups. Bayesian statistics do not permit to support one or the other hypothesis (alternative or null) ($BF_{10}$ = 0.77; $BF_{01}$ = 1.28).

**Mental flexibility. In second grade**, no effect of language group was found in correct responses ($F$ (2.138) = 0.03, $p$ = .96, $\eta p^2$ <0.01) or on median reaction times ($F$ (2.138) = 0.49, $p$ = .61, $\eta p^2$ <0.01). The absence of effect of group is confirmed by Bayesian statistics for correct responses ($BF_{10}$ = 0.07) and for median reaction times ($BF_{10}$ = 0.10).

**In fifth grade**, the results revealed a language group effect on correct responses ($F$ (2.86) = 4.76, $p$ = .01, $\eta p^2$ 0.09, $BF_{10}$ = 4.11) but not on median reaction times ($F$ (2.86) = 2.05, $p$ = .13, $\eta p^2$ 0.04, $BF_{10}$ = 0.50). Newman-Keuls post-hoc comparisons revealed that the English immersed group performed significantly better ($p$ < .01, $BF_{10}$ = 7.90) and the Dutch immersed group performed marginally better ($p$ = .050, $BF_{10}$ = 1.17) in terms of correct responses than the non-immersed group. No difference was found between the immersed groups ($p$ = 0.29, $BF_{10}$ = 0.47).

**Working memory. In second grade**, no effect of language group was found ($F$ (2, 138) = 0.93, $p$ = .39, $\eta p^2$ = 0.01). The absence of effect of group is confirmed by Bayesian statistics ($BF_{10}$ = 0.15).

**Table 6. BF resulting from Bayesian ANOVAs on alerting, cognitive flexibility, and working memory measures in Grade 5.**

| Variable | Model | BF10 | BF01 | error % |
|---|---|---|---|---|
| Alerting (RT) | Group effect (Im or NonIm) | 0.778 | 1.286 | 0.022 |
| Cognitive flexibility (RT) | Group effect (Im or NonIm) | 0.505 | 1.979 | 0.036 |
| Cognitive flexibility (CR) | Group effect (Im or NonIm) | **4.114** | 0.243 | 0.014 |
| Working memory (CR) | Group effect (Im or NonIm) | **2.474** | 0.404 | 0.029 |

*Note*. CR = Correct Response; RT = Reaction Times; $BF_{10}$ = Bayes factor for the alternative hypothesis vs. the null hypothesis; BF 01 = Bayes factor for the null hypothesis vs. factor for the alternative hypothesis.

**Table 7. Means (Standard Deviations) for the arithmetic task administered by grade.**

|  | Grade 2 | | | Grade 5 | | |
|---|---|---|---|---|---|---|
|  | ID Mean (SD) | IE Mean (SD) | NI Mean (SD) | ID Mean (SD) | IE Mean (SD) | NI Mean (SD) |
| TTR additions (CR) | 13.1 (3.1)* | 12.1 (3.9) | 13.0 (3.9) | 24.4 (3.2) | 21.5 (3.9) | 22.4 (3.8) |
| TTR subtractions (CR) | 11.4 (4.3) | 9.1 (3.8) | 10.8 (3.2) | 21.2 (3.4) | 17.8 (3.8) | 18.9 (3.9) |
| TTR multiplication (CR) |  |  |  | 20.1 (4.3) | 17.7 (3.6) | 18.9 (4.0) |
| TTR division (CR) |  |  |  | 17.1 (5.0) | 11.9 (4.6) | 13.3 (4.0) |
| TTR mixed (CR) |  |  |  | 20.5 (3.9) | 17.1 (3.1)* | 17.7 (3.4) |
| Composite Score (CR) [1] | 24.5 (6.7) | 21.2 (6.8) | 23.8 (6.3) | 103.4 (17.1) | 84.4 (18.2) | 91.2 (16.3) |

*1 missing data in the Dutch immersed group in second grade and 3 missing data in the English immersed group in fifth grade due to a problem during the passing.

[1] Composite score comprised additions and subtractions in grade 2 and all the calculations in grade 5. CR = Correct responses.

**In fifth grade**, an effect of language group was found ($F$ (2.81) = 4.09, $p$ = .02; $\eta p^2$ 0.08). The $BF_{10}$ factor for group effect is 2.47, which is interpreted as anecdotal evidence. Newman-Keuls post hoc comparisons and Bayesian comparisons support superior working memory performances in the Dutch immersed group in comparison with the non-immersed group in terms of the number of correct responses ($p$ = .01, $BF_{10}$ = 14.48). No difference was found between the English immersed group and the non-immersed group ($p$ = .12, $BF_{10}$ = 0.67) or between the immersed groups ($p$ = .20, $BF_{10}$ = 0.48).

**Arithmetic achievement.** Concerning the arithmetic achievement evaluated with the Tempo Test Rekenen [41], ANOVAs (Table 7) revealed no difference between the groups in second grade for additions ($F$ (2.137) = 0.91, $p$ = .40, $\eta p^2$ 0.01, $BF_{01}$ = 6.66) but a difference was found for subtractions ($F$ (2.137), $p$ = .01, $\eta p^2$ 0.06, $BF_{10}$ = 3.49). Planned comparisons showed that English immersed children performed worse than Dutch immersed children ($t$ = 2.9, $p <$ .01; $BF_{10}$ = 5.89) and the non-immersed children ($t$ = 2.1, $p$ = .03; $BF_{10}$ = 2.26). Note however that this last result should be interpreted with caution. Indeed, if we apply a correction for multiple t-test correction ($p <$ .016), the difference became insignificant. No significant difference was found between non-immersed and Dutch immersed children ($t$ = 0.84, $p$ = .40; $BF_{10}$ = 0.29).

In fifth grade, a group effect was found for additions ($F$ (2.86) = 4.69, $p$ = .01; $\eta p^2$ 0.09, $BF_{10}$ = 3.1), for subtractions ($F$ (2.86) = 6.26, $p <$ .01; $\eta p^2$ 0.12, $BF_{10}$ = 12.8), divisions ($F$ (2.86) = 10.29, $p <$ .001; $\eta p^2$ 0.19, $BF_{10}$ = 252.5) and mixed calculations ($F$ (2.86) = 7.55, $p <$ .001; $\eta p^2$ 0.15, $BF_{10}$ = 33.7). For multiplications, no significant group effect was found ($F$ (2.86) = 2.72, $p$ = .07; $\eta p^2$ 0.05, $BF_{10}$ = 0.8). A group effect was also found with the composite score ($F$ (2.86) = 9.23, $p <$ .001; $\eta p^2$ 0.17, $BF_{10}$ = 116.2). Planned comparisons showed that Dutch immersed children performed better in all the calculations (except multiplications) than their control pairs and the English immersed children. No differences were found between English immersed and non-immersed children.

## Discussion

The aim of the present study was to explore if the second language learned (Dutch or English) and the time spent to learn a second language (two or five years) in an early immersion school program could influence outcomes of studies investigating executive and academic performances in immersed children. We evaluated children after 2 years (second grade) and 5 years (fifth grade) of early bilingual education experience. Within the two grade groups, we compared French-speaking children learning Dutch and English as second-language with a French-speaking monolingual group on tasks evaluating alerting, cognitive flexibility, and

working memory. We also evaluated the arithmetic abilities of these different groups to determine if the second language learned and the time spent in immersion could also influence the arithmetic performances in CLIL context.

## Advantages in the function of the second language learned—English vs. Dutch–and the time spent in immersion

After 2 years of L2 immersion, no advantage seems to emerge for the English or the Dutch groups in comparison with the non-immersed group on the different tasks administered. Concerning alerting and cognitive flexibility, the results are in line with those of Gillet et al. [5] for French-speaking children immersed in Dutch and with those of Barbu et al. [29] for French-speaking children immersed in English showing no advantage on the same tasks as ours evaluating alerting and cognitive flexibility after 2 years of immersion. Concerning working memory, to our knowledge, only Gillet et al. [5] evaluated this function with the same task as ours in immersed children. They no longer find an advantage for French-speaking children immersed in Dutch after 2 years of immersion.

After 5 years of immersion, advantages emerge but differ in function of the second language learned except for alerting, where no difference was found between the three groups. Concerning cognitive flexibility, Bayesian statistics showed moderate evidence for an advantage in children immersed in English ($BF_{10}$ = 7.90) and anecdotal evidence for an advantage in children immersed in Dutch ($BF_{10}$ = 1.17) in comparison with non-immersed children. Concerning working memory, Bayesian statistics showed strong evidence for an advantage in children immersed in Dutch ($BF_{10}$ = 14.48) while no evidence in English immersed children ($BF_{10}$ = 0.67) in comparison with non-immersed children. No other study with Dutch or English immersed children evaluated alerting, cognitive flexibility, and working memory using the same tasks as in the present study in fifth grade.

Considering the outcomes of the different studies that have used the same tasks to assess alerting, cognitive flexibility and working memory, at different moments of primary schooling, in French-speaking children immersed in English or in Dutch, it seems that, if a cognitive advantage is demonstrated, the moment of appearance of the cognitive advantages, as well as the specific cognitive function(s) enhanced, vary in function of the language learned. Moreover, once the advantage appears, it may not be sustainable. *In English immersed children*, advantage in selective auditory attention for example appears in first grade [3], not in second grade, reappear in third grade [2] and is not found later in CLIL schooling. The present study showed an advantage in cognitive flexibility only later in the schooling (fifth grade) confirming the outcomes of a previous study in Dutch CLIL context, that found no advantages in first, second, or third grades while advantages were found in sixth grade in cognitive flexibility and working memory [5].

To resume, concerning *English immersion*, the results seem to show that cognitive advantages are fluctuating over time with a "peak" at mid-program (in third grade, all the functions were higher in immersed children apart from inhibition). Concerning *Dutch immersion*, it seems that no advantages are observed at the beginning of the program (first three grades) but that some advantages emerge at the end of it (in fifth and sixth grade).

Globally, during primary schooling, *alerting* seems not to be enhanced by CLIL context in second and fifth grade immersed children. This function would not be more solicited in a CLIL context, or this solicitation is not sufficient to enhance this function durably. Future studies may use slightly more demanding tasks than the detection of simple visual stimuli such as an odd/even number judgement for example. The use of these more complex reaction time tasks would allow for a better characterization of the specificity the reaction time differences in tasks involving cognitive flexibility.

Contrariwise, the CLIL context seems to train *cognitive flexibility* more than the traditional school context at some moments. On the one hand, switching from one language to another in the CLIL context requires cognitive flexibility abilities. This behaviour has been proposed as an explanation of the cognitive flexibility advantage in bilinguals [15,16]. On the other hand, children also have to be more focused on the L2 particularities when listening to the teacher's instructions and have frequently at the same time to perform another task like writing, for example, and to switch between the two tasks. Cognitive flexibility is, consequently, more trained and, in turn, enhanced in the CLIL context. As a reminder, differences were observed between the two language groups (English and Dutch) for cognitive flexibility. That is to say, a higher likelihood of the advantage in favour of children learning English as L2 ($BF_{10}$ = 7.90 for English and $BF_{10}$ = 1.17 for Dutch) and earlier emergence of the advantage in the English CLIL schooling [2]. We hypothesized that these differences could be due to the higher proximity between English and French languages than between Dutch and French languages. Another and not exclusive explanation could come from the incidental learning of English out of the school context. English, more than Dutch, for French-speaking children is used in many authentic contexts and integrated into many people's daily activities, such as listening to music, using the internet or social media, or gaming. For example, De Wilde and Eyckman [14] investigated incidental English language acquisition of 10 to 12-year-old Dutch children who did not receive any formal English instruction. They measured children's English proficiency (receptive lexical test and a test measuring listening skills, speaking skills, reading skills, and writing skills). Results showed that a significant proportion of the children already performed tasks at the A2 level (i.e., Elementary English level). This study confirms that children learn English from the input they receive through different media (especially gaming and computer use). The study also revealed a high positive attitude towards English. The authors showed that the most beneficial types of input were gaming, use of social media, and speaking, and highlighted that all these inputs are interactive, multimodal, and involving language production. In this context, the children learning English in an immersion school could be more and earlier exposed to their L2 than children learning Dutch in immersion and, therefore, better train their abilities to switch from one linguistic context to the other. This training would affect their cognitive flexibility abilities earlier in their development. The non-immersed children seem, however, to catch their English pairs up at the end of the schooling [6] maybe as they can no longer evolve because of a ceiling in the development curve. Future studies should further investigate the time children spend playing video gaming, watching movies, and using social media and their computer, and in which language these activities are made (French or English vs. Dutch). Moreover, it has been shown that students are more motivated to learn English than one of the other national languages in Belgium [18], which could also influence the outcomes.

Concerning the *working memory* task, only the Dutch immersed group showed an advantage. This could be due to the specific structure of the Dutch language, which differs from the French language. At the sentence level, as mentioned in the introduction, the underlying structure in Dutch is different from those of French and English that are more similar (sentence with a subject-verb-object structure). In Dutch, some verbal forms are placed at the end of the sentence as the infinitives (e.g., to say The kids have *to eat* vegetables in English, Les enfants doivent *manger* des légumes in French, it is said in Dutch De kinderen moeten groenten *eten* which would literally correspond to the children have vegetables to eat in English) and past participles (e.g., to say, I *have eaten* vegetables in English, or J'*ai mangé* des légumes in French, it is said in Dutch Ik heb groenten *gegeten* which would give in English I have vegetables eaten). In sub-clauses, all verbal forms are rejected at the end of the sentence (e.g., to say, I see the bord that the spin *attacks* in English, or Je vois l'oiseau qui *attaque* l'araignée in French, it

is said in Dutch Ik zie de vogel die de spin *aanvalt*, which in English would give I see the bord that the spin attacks). Because the verb can be regarded as the head of the predicate, Dutch structure is said to be head-final (the head of the phrase–that is to say, the verb–is in the final position), whereas the English and French structures are head-initial (the head of the phrase–that is to say, the verb–is in initial position). This particularity could involve differently the working memory (WM) abilities in monolinguals according to the syntactic structure of their language [13]. We hypothesized that it could also be the case, and maybe even more, when this structure concerns the second language learned. The load in working memory when trying to understand an oral sentence in L2 in which the head of the phrase is at the end could be particularly high for children with a head-initial L1 language. This recurring exercise could consequently improve working memory abilities.

Moreover, another explanation that could be invoked could be related to the difference in the way of expressing the numbers in French and in Dutch, in which the order is different, e.g., in Dutch, 45 is spoken vijfenveertig, literally five and forty. Indeed, Bahnmueller, Göbel, Pixner, Dresen, & Moeller [49] have reported that Arabic number processing could be moderated by linguistic specificities, such as the inversion property of number words. Despite the fact that some studies evaluating the influence of linguistic properties on processing place-value information have shown that number word inversion leads to additional processing costs in different numerical tasks [e.g., multi-digit addition: 27], some authors have suggested that inversion could actually be advantageous for the retrieval of the correct solution of some calculations [for further information, see 49]. In the context of Dutch CLIL, the learning and the manipulation of two different number structures, like immersed children have to do as soon as in the third kindergarten, can have an early impact on the understanding of the place-value and, in turn, in the calculation. The early place-value understanding was indeed shown to predict later arithmetic abilities [50]. Moreover, this way of expressing the numbers and the availability of two systems (number word inversion or not) in children's mind, likely requires more working memory abilities. For example, in the Dutch-French pair of languages, transcoding from one notation to another is more challenging given the inconsistency between verbal number words and Arabic numbers [27].

Arguments against a cognitive advantage of bilingualism.

It has been shown that there is publication bias toward studies showing a bilingual advantage. Several authors suggested that this advantage does not exist or that it is really small and/or task-dependent [51–54]. Concerning the advantage of cognitive flexibility and inhibitory control observed in developmental studies, the results have rarely been replicated in a reliable manner [55] and, when these studies are taken together, it seems that "no single task or dependent variable has consistently been found to index an early bilingual advantage in cognitive inhibition or flexibility" [55, p. 9]. In the context of CLIL (elementary school) studies, Gillet and Poncelet [56] did a comprehensive review of the studies conducted between 2008 and 2020 and showed that only eight among eighteen studies showed advantages in one or more functions among immersed children, which represents less than the half of the studies conducted. They also evidenced that there was a high heterogeneity among the studies in terms of time spent in immersion or languages involved. In the present study, we used standardized tasks that had already shown cognitive advantages in English immersed children. Moreover, as performance for alertness was comparable between the groups, the advantage found in cognitive flexibility and working memory in the present study may appear to be specific. However, the field is confronted to a poorly defined theoretical background and, consequently, to a lack of congruency in the tasks used to assess cognitive development. Using several tasks to assess one cognitive function combined with a latent variable approach could also help to characterize in a more robust manner which mechanisms are affected by second language learning and the CLIL context [57].

Furthermore, larger sample sizes, even if perhaps difficult and costly to recruit, will also allow for a more reliable characterization of the cognitive impact of bilingualism [58]. As mentioned by Brysbaert [58], "Simulations indicate that little research with sample sizes lower than 100 participants per group provides a picture of enough resolution to draw firm conclusions."

## The impact of time spent in immersion and languages at stake on arithmetic abilities in CLIL context

Here we found, in second grade, a disadvantage for English immersed children in some calculations. However, as mentioned in the result section, this result would be interpreted with caution. In fifth grade, we found a superiority for the Dutch immersed group in arithmetic abilities, yet the disadvantage for English immersed children in second grade is no longer observed. Globally, the results suggest a role of the time spent in CLIL context and the languages at stake on arithmetic abilities as advantages appear only in Dutch CLIL context and only at the end of the elementary CLIL program.

In second grade, no difference was found between the groups for additions. However, for subtractions, results suggest that the English immersed children performed slightly worse than the Dutch language group. These results are surprising given that most of the studies showed that the systems in which the order of number words match with the order of digits in the Arabic number system (like in English and French languages) were found to be advantageous for monolinguals children's numerical development [e.g., 59]. However, Barbu and al. [29] found a disadvantage of their French-speaking children immersed in English on arithmetic's after 2 years of immersion. This finding was explained by the authors in terms of an increased cognitive load engendered by the constant treatment of L2 information that is less mastered than the L1 at the beginning of the schooling. Yet, if true, this will also be the case for Dutch immersed children. In Dutch, the use of different word numbers, with an inverse ten-unit order, should consequently even more slowdown the processing because of interference. However, as we mentioned it above, the early confrontation to different systems could enhance comprehension of the unit-tens concepts and, in turn, compensate this interference. Or maybe facilitate the processing of some calculations, e.g. in which the units have to be process first, as some authors had already emit the hypothesis [59]. The task used in the present study is not sufficient to further assess this hypothesis. Future studies should investigate this possibility.

In fifth grade, Dutch immersed children outperformed the English and the non-immersed groups. Note that there was no difference between the English and the non-immersed groups. The advantage for the Dutch immersed children could be related to their higher skills in working memory. Another study has also shown that the working memory (visuospatial) was superior in bilingual pre-schoolers (4–5 years old) and linked with calculation performances [25]. However, they did not study the couple English or Dutch-French language and did evaluate the visuospatial working memory and not verbal working memory. Working memory is crucial for complex cognitive activity, such as mathematical learning [60]. Indeed, different studies suggest that working memory abilities affect the ability to remember the lexical elements, their sequence, and to manipulate this sequence in different formats (e.g., transcoding) and calculations [e.g., 61–64]. Another–but not exclusive–explanation for the enhanced calculation abilities in French-speaking children learning Dutch could be that learning arithmetic in two different number-word systems influences the abstraction of the concept of number positively and, in turn, could help for the execution of calculation operations. As we mentioned above, this early confrontation to two different number structures can have an early impact on the understanding of the place-value and, in turn, on calculation as early place-value understanding was shown to predict later arithmetic abilities [57].

Other factors could also be mentioned to explain the differences found between the languages at stake, such as the method of teaching mathematics that was not controlled in the present study. Indeed, teaching content in a language that children do not master could induce adaptations in pedagogic practices (the teacher could use material that is more concrete, propose more manipulations and so on). These modifications in teaching practices could have an impact on attentional and executive functioning but also directly on school achievement. Moreover, children came from different schools that induce that the pedagogical practices also could differ. Nevertheless, we excluded schools known to use particular curricula as "active pedagogy".

Finally, future studies should evaluate the cognitive development with a longitudinal approach to assure that the children have the same cognitive level before immersion with a greater sample. Individual differences as motivation, learning strategies, personality traits, metalinguistic awareness, emotions, and beliefs about L2 learning and mathematics or political context could affect the cognitive and academic performances of children [65]. De Smet, Mettewie, Galand, Hiligsmann, and Van Mensel [66], for example, showed that children in fifth grade attending English CLIL classes were less anxious and experienced more enjoyment in their L2 class then the Dutch CLIL children. Future studies should also try to consider these factors to evaluate how they affect cognitive performances.

## Conclusion

To conclude, our study comparing children following a bilingual education in English vs. Dutch for 2 or 5 years shows that 2 years of early bilingual education seems not to bring evidence for an advantage on executive functions, as measured by our tasks. However, bilingual education experience seems to bring some cognitive advantages after 5 years. We used some executive tasks already known to show an advantage [1,2,5] in French-speaking children learning English or Dutch as L2. In fifth grade, results differ in function of the L2 learned. While immersion seems to lead to an advantage in cognitive flexibility at the end of the primary scholarship for both languages, learning Dutch seems to contribute to an additional advantage in working memory that does not appear in English. This advantage in EF, associated with the nature of the second language number-word system (that is different in Dutch), could lead to better performances in arithmetic in Dutch. This has to be confirmed by future works. Other studies should now try to replicate the results with the same tasks but with a longitudinal design and controlling more the extra scholar activities that could have an influence on L2 learning as well as on attentional and executive performances (music listening, video games in an L2, and the language in which the activities are made...).

## Acknowledgments

We thank Sarah Tellatin and Audrey Gonzalez for their contribution to data collection.

## Author Contributions

**Conceptualization:** Sophie Gillet, Martine Poncelet.

**Data curation:** Sophie Gillet, Cristina Barbu.

**Formal analysis:** Sophie Gillet, Martine Poncelet.

**Methodology:** Sophie Gillet, Martine Poncelet.

**Supervision:** Martine Poncelet.

**Writing – original draft:** Sophie Gillet, Cristina Barbu, Martine Poncelet.

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
