## [Decision Letter · Decision Letter 0]

21 Apr 2021

PONE-D-21-09083

Early bilingual immersion school program and cognitive development in French-speaking children: Effect of the second language learned (English vs. Dutch) and exposition duration (2 vs. 5 years)

PLOS ONE

Dear Dr. Gillet,

Thank you for submitting your manuscript to PLOS ONE. After careful consideration, we feel that it has merit but does not fully meet PLOS ONE’s publication criteria as it currently stands. Therefore, we invite you to submit a revised version of the manuscript that addresses the points raised during the review process.

We look forward to receiving your revised manuscript.

Kind regards,

Dr Roberto Filippi

Academic Editor

PLOS ONE

Journal Requirements:

We note that you have stated that you will provide repository information for your data at acceptance. Should your manuscript be accepted for publication, we will hold it until you provide the relevant accession numbers or DOIs necessary to access your data. If you wish to make changes to your Data Availability statement, please describe these changes in your cover letter and we will update your Data Availability statement to reflect the information you provide.

Reviewers' comments:

Reviewer's Responses to Questions

**Comments to the Author**

1. Is the manuscript technically sound, and do the data support the conclusions?

Reviewer #1: Yes

Reviewer #2: Partly

2. Has the statistical analysis been performed appropriately and rigorously? 

Reviewer #1: Yes

Reviewer #2: No

3. Have the authors made all data underlying the findings in their manuscript fully available?

Reviewer #1: No

Reviewer #2: Yes

4. Is the manuscript presented in an intelligible fashion and written in standard English?

Reviewer #1: Yes

Reviewer #2: Yes

5. Review Comments to the Author

Reviewer #1: This study tested for differences in three components of executive functioning (alerting, cognitive flexibility, and working memory) and arithmetic abilities in children (either 2nd or 5th grade) in three language groups: English immersed, Dutch immersed, and non-immersed French monolinguals.

No differences between the three language groups were obtained in second grade, but in fifth grade the two immersed groups outperformed the monolingual controls on the cognitive flexibility task (but did not differ from each other). Only the Dutch immersed group outperformed the monolingual control in the working memory task. With respect to arithmetic, the Dutch immersed group outperformed the monolingual controls who, in turn, were better than the English immersed group.

More should be said about how these groups were identified. I assume that each language group was sampled from a different school or different sets of schools. Although the groups are matched on SES and general fluid intelligence (that’s great!) there exists the possibility that the differences that eventually emerge in 5th grade are “school” effects, not “language use” effects. Regardless of the language(s) of instruction there are good schools and not-so-good schools in local communities that vary in the richness of the extracurricular experiences that they offer. Because those differences emerge in the fifth grade where the sample size is only 30 children, one wonders what the chances are of reproducing say the same pattern of differences in mental flexibility if one replicated the original study, but with a new set of schools. This problem is more vexing when predictions are derived from underdeveloped theories and inconsistent earlier results. For example, are the interesting conjectures regarding the obtained differences between English and Dutch immersion really caused by language similarity or might they be just as likely due to school differences or the riskiness associated with small samples sizes (N=30)? I think current practice encourages psychologists to explain every significant difference that is observed and that we often overfit the data – a point effectively addressed by Gullifer and Titone in a new JEP:General article.

Another detail that might be worth mentioning is the conditions of testing for each group/school and whether the same experimenter did the testing and was it always done in French?

Another issue that deserves more discussion is the decision to use only one task to derive a measure of each of the three targeted EFs (alerting, mental flexibility, and working memory capacity). Concerns about the existence of a domain-free components of inhibitory control are becoming acute (Paap, Anders-Jefferson, Zimiga, Mason, & Mikulinsky, 2020) because the interference control in the flanker, Simon, and spatial Stroop tasks appears to be task specific. Alarmingly, back in 2010 Salthouse reported that the letter and arrow instantiations of the flanker task do not correlate. Paap, et al. (2019) reported that two versions of the Simon task and a spatial Stroop task cohered into a latent variable, but that an arrow-version of the flanker task did not load on this variable. A study by Rey-Mermet, Gade, and Oberauer (2018) used six tasks assumed to reflect Inhibition of Prepotent Responses and five assumed to reflect Resistant to Distraction. Bayesian hypothesis testing showed that the data provide ambiguous evidence as to whether there is one inhibition factor or two; or, if two, whether they are correlated or orthogonal. They conclude that nonverbal tests used to assess “inhibition” do not measure a common, underlying construct but instead measure the highly task-specific ability to resolve the interference arising in each task. For them the “... inevitable implication is that studies using a single laboratory paradigm for assessing or investigating inhibition do not warrant generalization beyond the specific paradigm studied” (p. 515). Similarly, Paap, et al. (2020) recommended that we should stop evaluating the consequences of bilingualism (or other special experiences) on EF by using single tasks, especially the flanker task, because these reflect mostly task-specific control mechanisms. Indeed, one reason why it may be so difficult to consistently produce significant differences between types of bilinguals or between bilinguals and monolinguals is that bilingual language control is encapsulated within the language processing system (Paap et al., 2019) and, consequently, is different from the task-specific mechanisms used in the common measures of EF. Blanco-Elorrieta and Pylkkanen (2018) have made a similar argument about switching. They reviewed a body of work showing that when bilinguals switch languages voluntarily, both the behavioral switch costs and the activation of brain regions associated with cognitive control are greatly reduced or eliminated. This pattern suggests that switching languages is not inherently effortful, does not usually require top-down control, and therefore bilingual advantages in general switching costs may be limited to bilinguals who frequently switch languages based on unpredictable external constraints. To be fair, much of the discussion in this paragraph has grown from tasks assumed to measure inhibition and the current study did not include a measure of inhibitory control! Nonetheless, as introduced at the beginning of the paragraph the authors heavily invest in the assumption that domain general tests of specific cognitive abilities can be measured with single tasks. Using a latent variable approach would be superior.

The authors do not review the more general literature on the bilingual advantage in EF hypothesis for children and I would be interested in how they view the smaller subset of studies investigating the effects of immersion. Here’s my quick review of the general results with kids. Paap (2019) reported that only 3 of the 30 comparisons using children in the range of 6 to 15 years old produced significant bilingual advantages in nonverbal interference scores (assumed by many to measure inhibitory control) and that the mean effect size was +2.2 ms (95% CI: -7.9, +12.2). Furthermore, very large‐scale studies with highly proficient bilingual children living in language communities where language switching occurs all the time have shown no bilingual advantages in non‐verbal interference tasks (Antón, Duñabeitia, Estévez, Hernández, Castillo, Fuentes, Davidson, & Carreiras; Duñabeitia, Hernández , Antón, Macizo, Estévez, Fuentes, & Carreiras, 2014; Gathercole, Thomas, Kennedy, Prys, Young, Vinas-Guasch, ... Jones, 2014). Bialystok (2017) dismisses these results because they “examine an unusually large age range without convincing control over the role of age in performance” (p. 238) but all of these studies analyze the results in separate and narrow age bands with no hint that age or years of bilingual experience matters. Adding more weight to the conclusion that bilingual advantages do not consistently or significantly occur in children is the recent meta-analysis reported by Gennerud, ten Braak, Reikeras, Donolato, and Melby-Lervåg (2020) showing an overall effect size of g = 0.06 (and indications of publication bias) based on 583 effect sizes.

Reviewer #2: Introduction

Dutch-French/English-French comparison (starting page 5, line 102 and ending at page 7, line 150) - I find the outline of the linguistic comparisons and potential links to non-verbal cognition quite lengthy, unfocused, and confusing. I’d encourage the authors to rewrite this entire section to first outline differences and then explain how these may lead to different cognitive benefits.

Paragraph starting page 7, line 160 - this could be the end of the previous paragraph

Paragraph starting page 7, line 163 - what is the interpretation of these differences (if available)

Paragraph starting page 8, line 174 - I suggest moving this paragraph to the end of this section (essentially below the next one) and expanding on it a little - it is not entirely clear what this adds here above and beyond what has already been discussed. I also believe it should be acknowledged at this point that findings in this area are mixed.

Page 10, line 239 - I find the phrasing here rather weak given that literature delivers no clear direction. I’d argue that rather than it being ‘interesting to see what happens’, the study rather aims to provide more information as to whether alerting is affected by immersion or not.

Page 11, line 250 - Again, I don’t really think the phrasing here is ideal in terms of providing clear aims. This section also raises the question why these particular year groups were chosen. Again, I’d encourage the authors to provide clearer aims/hypotheses and to avoid the rather ambiguous notion of something being interesting to look at.

Page 11, line 264 - As noted before, the notion of something being ‘interesting’ doesn’t sit well with me - are the authors trying to establish at which point during development such benefits may occur? If so, I believe this should be clearly stated.

Page 12, line 269 - I would remove the first sentence, the second says something very similar but acknowledges that alternative findings would be possible

Method

Page 13, line 298 - socioeconomic?

Page 14, Table 2 - where ES=0.00, should it not be ES<0.01?

Results

General - I am wondering why median and not mean reaction time was analysed here, is this common practice for this task

Page 18, line 419 - if at all, I would call this marginally non-significant, especially given the Bayesian statistics - in general, it appears that the data cannot provide consistent support for alerting differences in either case

Paragraph starting page 19 , line 426 - ‘=‘ signs missing for partial eta squared and once again I find < 0.01 more meaningful than = 0.00

Page 19, line 445 - I recommend exchanging the word ‘superiority’ with a slightly more balanced/precise one, such as ‘performance advantage’ or even ‘superior WM performance’

Page 21, line 477 - it is questionable whether the comparison between English- and non-immerse children should be considered significant at p = .03 or would warrant correction which means it would not be, in either case I suggest the authors include a note that this should be interpreted with caution

Page 22, line 485 - where are these comparisons reported?

Correlations - I am not convinced that these analyses are very meaningful as they are presented, just because one correlation is significant and another is not does not mean they are significantly different (.14 and .46 are, for example, not according to a Fisher’s r-to-z transformation). I suggest the authors consider what they are really seeking to evaluate here and to reconsider the analyses accordingly. Possibly multiple regressions with dummy coding for groups would yield clearer results.

Discussion

Page 25, paragraph 1 - I recommend rewriting and shortening this paragraph as there is quite a bit of repetition in it and it is not very easy to follow

Page 26, paragraph 1 - it could be more clear why the authors consider cognitive demand to be a potential confound

Page 27 - this could be shorter - in general I feel like there is quite a bit of repetition in the discussion, the section could be more concise overall

I also recommend updates in terms of a cautious evaluation of the data as appropriate and as recommended in the commentary on the results section

6. PLOS authors have the option to publish the peer review history of their article (what does this mean?). If published, this will include your full peer review and any attached files.

Reviewer #1: **Yes: **Ken Paap

Reviewer #2: **Yes: **Dr Julia Ouzia

---

## [Author Response · Author response to Decision Letter 0]

16 Jul 2021

Response to the Reviewers : Early bilingual immersion school program and cognitive development in French-speaking children: Effect of the second language learned (English vs. Dutch) and exposition duration (2 vs. 5 years) submitted to PLOS ONE

First of all, we thank the reviewers for their interest in our work, the time they spent reviewing it and their valuable comments.. 

Please submit your revised manuscript by Jun 05 2021 11:59PM. 

We note that you have stated that you will provide repository information for your data at acceptance. Should your manuscript be accepted for publication, we will hold it until you provide the relevant accession numbers or DOIs necessary to access your data. If you wish to make changes to your Data Availability statement, please describe these changes in your cover letter and we will update your Data Availability statement to reflect the information you provide.

Comments to the Author

1. Is the manuscript technically sound, and do the data support the conclusions?

Reviewer #1: Yes

Reviewer #2: Partly

2. Has the statistical analysis been performed appropriately and rigorously? 

Reviewer #1: Yes

Reviewer #2: No

3. Have the authors made all data underlying the findings in their manuscript fully available?

Reviewer #1: No

Reviewer #2: Yes

4. Is the manuscript presented in an intelligible fashion and written in standard English?

Reviewer #1: Yes

Reviewer #2: Yes

5. Review Comments to the Author

Reviewer #1: 

This study tested for differences in three components of executive functioning (alerting, cognitive flexibility, and working memory) and arithmetic abilities in children (either 2nd or 5th grade) in three language groups: English immersed, Dutch immersed, and non-immersed French monolinguals.

No differences between the three language groups were obtained in second grade, but in fifth grade the two immersed groups outperformed the monolingual controls on the cognitive flexibility task (but did not differ from each other). Only the Dutch immersed group outperformed the monolingual control in the working memory task. With respect to arithmetic, the Dutch immersed group outperformed the monolingual controls who, in turn, were better than the English immersed group.

More should be said about how these groups were identified. I assume that each language group was sampled from a different school or different sets of schools. Although the groups are matched on SES and general fluid intelligence (that’s great!) there exists the possibility that the differences that eventually emerge in 5th grade are “school” effects, not “language use” effects. Regardless of the language(s) of instruction there are good schools and not-so-good schools in local communities that vary in the richness of the extracurricular experiences that they offer. Because those differences emerge in the fifth grade where the sample size is only 30 children, one wonders what the chances are of reproducing say the same pattern of differences in mental flexibility if one replicated the original study, but with a new set of schools. This problem is more vexing when predictions are derived from underdeveloped theories and inconsistent earlier results. For example, are the interesting conjectures regarding the obtained differences between English and Dutch immersion really caused by language similarity or might they be just as likely due to school differences or the riskiness associated with small samples sizes (N=30)? I think current practice encourages psychologists to explain every significant difference that is observed and that we often overfit the data – a point effectively addressed by Gullifer and Titone in a new JEP: General article.

Response: 

In Belgium, most schools use the same type of pedagogy and we have discarded schools known to practice particular pedagogies such as “active pedagogy” (e.g. “Tools of the Mind” pedagogy). Children from each language group came from four to five different schools. None of these schools was known to practice particular pedagogies. We have added this information in the manuscript. 

The sample size can indeed constitute a problem. However, the Bayesian factors for the group effect are 4.11 for cognitive flexibility (BF10 = 7.90 for the comparison of children immersed in English and non-immersed children) and 2.47 for working memory (BF10 = 14.48 for the comparison of children immersed in Dutch and non-immersed children). These factors sustain inferential evidence (even if they constitute a moderate evidence in favour of a group difference). Moreover, the present study replicated the outcomes of Gillet, Barbu, and Poncelet (2020) with the same tasks with another sample. We have added the issue of the risk associated with small samples as well as the question of the underdeveloped theories in the Discussion section (P30, line 587-597 and line 700-703). 

Another detail that might be worth mentioning is the conditions of testing for each group/school and whether the same experimenter did the testing and was it always done in French?

Response: The children were all tested individually in their school in the morning. There were two experimenters for the children in fifth grade, and six experimenters for the children in second grade. The experimenters had to test the same number of immersed and non-immersed children. French (mother tongue of the children) was indeed the language of testing for all the children. We added this information in the Method part. 

Another issue that deserves more discussion is the decision to use only one task to derive a measure of each of the three targeted EFs (alerting, mental flexibility, and working memory capacity). Concerns about the existence of a domain-free components of inhibitory control are becoming acute (Paap, Anders-Jefferson, Zimiga, Mason, & Mikulinsky, 2020) because the interference control in the flanker, Simon, and spatial Stroop tasks appears to be task specific. Alarmingly, back in 2010 Salthouse reported that the letter and arrow instantiations of the flanker task do not correlate. Paap, et al. (2019) reported that two versions of the Simon task and a spatial Stroop task cohered into a latent variable, but that an arrow-version of the flanker task did not load on this variable. A study by Rey-Mermet, Gade, and Oberauer (2018) used six tasks assumed to reflect Inhibition of Prepotent Responses and five assumed to reflect Resistant to Distraction. Bayesian hypothesis testing showed that the data provide ambiguous evidence as to whether there is one inhibition factor or two; or, if two, whether they are correlated or orthogonal. They conclude that nonverbal tests used to assess “inhibition” do not measure a common, underlying construct but instead measure the highly task-specific ability to resolve the interference arising in each task. For them the “... inevitable implication is that studies using a single laboratory paradigm for assessing or investigating inhibition do not warrant generalization beyond the specific paradigm studied” (p. 515). Similarly, Paap, et al. (2020) recommended that we should stop evaluating the consequences of bilingualism (or other special experiences) on EF by using single tasks, especially the flanker task, because these reflect mostly task-specific control mechanisms. Indeed, one reason why it may be so difficult to consistently produce significant differences between types of bilinguals or between bilinguals and monolinguals is that bilingual language control is encapsulated within the language processing system (Paap et al., 2019) and, consequently, is different from the task-specific mechanisms used in the common measures of EF. Blanco-Elorrieta and Pylkkanen (2018) have made a similar argument about switching. They reviewed a body of work showing that when bilinguals switch languages voluntarily, both the behavioral switch costs and the activation of brain regions associated with cognitive control are greatly reduced or eliminated. This pattern suggests that switching languages is not inherently effortful, does not usually require top-down control, and therefore bilingual advantages in general switching costs may be limited to bilinguals who frequently switch languages based on unpredictable external constraints. To be fair, much of the discussion in this paragraph has grown from tasks assumed to measure inhibition and the current study did not include a measure of inhibitory control! Nonetheless, as introduced at the beginning of the paragraph the authors heavily invest in the assumption that domain general tests of specific cognitive abilities can be measured with single tasks. Using a latent variable approach would be superior. 

Response: We totally agree that this kind of study should eventually be conducted. However, in the present study, our main objective was to replicate the results found by previous studies using the same tasks. Therefore, we used the same tasks as those previously used in studies showing an advantage. These tasks come from a standardized battery commonly used in clinic settings and based on Sturm’s attention model [Sturm, Fimm, Cantagallo, Cremel, North., Passadori,., et al., 2002; Computerized training of specific attention deficits in stroke and traumatic brain-injured patients : A multicentric efficacy study. In M. Leclercq & P. Zimmermann (Eds.), Applied Neuropsychology of Attention (pp. 365-380). London : Psychology Press.] We agree that it would have been more informative to use different tasks to evaluate each function in regard to Paap’s papers on the question. As we used different tasks that are supposed to assess several cognitive functions, we could not use too much tasks per function because it consequently takes a lot of school time of the children. Moreover, in addition to the effect of learning a second language, we also aimed at assessing the effect of the CLIL context on attentional and executive components. Thus we evaluated not only bilingual language control that would be encapsulated within the language processing system, but also the impact of growing in an environment wherein the children have to learn something in particularly demanding conditions. Indeed, at the beginning of the CLIL schooling, children understand nothing of what is said by the teacher and have to switch frequently in function of the contexts. Nevertheless, we added that it would be necessary to modify the approach by using the latent variable approach to better understand which mechanism(s) is (are) susceptible to be implicated in second language learning and CLIL context the Discussion section. 

The authors do not review the more general literature on the bilingual advantage in EF hypothesis for children and I would be interested in how they view the smaller subset of studies investigating the effects of immersion. Here’s my quick review of the general results with kids. Paap (2019) reported that only 3 of the 30 comparisons using children in the range of 6 to 15 years old produced significant bilingual advantages in nonverbal interference scores (assumed by many to measure inhibitory control) and that the mean effect size was +2.2 ms (95% CI: -7.9, +12.2). Furthermore, very large‐scale studies with highly proficient bilingual children living in language communities where language switching occurs all the time have shown no bilingual advantages in non‐verbal interference tasks (Antón, Duñabeitia, Estévez, Hernández, Castillo, Fuentes, Davidson, & Carreiras; Duñabeitia, Hernández , Antón, Macizo, Estévez, Fuentes, & Carreiras, 2014; Gathercole, Thomas, Kennedy, Prys, Young, Vinas-Guasch, ... Jones, 2014). Bialystok (2017) dismisses these results because they “examine an unusually large age range without convincing control over the role of age in performance” (p. 238) but all of these studies analyze the results in separate and narrow age bands with no hint that age or years of bilingual experience matters. Adding more weight to the conclusion that bilingual advantages do not consistently or significantly occur in children is the recent meta-analysis reported by Gennerud, ten Braak, Reikeras, Donolato, and Melby-Lervåg (2020) showing an overall effect size of g = 0.06 (and indications of publication bias) based on 583 effect sizes.

Response: thank you for this remark and this very complete review. We added some of these references to our discussion. We also carried out a comprehensive review of the CLIL effect on attentional and executive functions at the primary level (Gillet & Poncelet, submitted) and observed, as in early bilingualism, inconsistent results. Only 10 out of 18 studies showed AEF advantages although some studies used highly similar tasks and evaluated highly homogeneous conditions of learning. We therefore wanted to explore the reasons of this inconsistency in the outcomes. One of these reasons could be the second language learned in CLIL in interaction with the time spent in immersion. We therefore conducted the present study in order to determine if the second language learned could be a potential significant factor of inconsistency. 

Reviewer #2: 

Introduction

Dutch-French/English-French comparison (starting page 5, line 102 and ending at page 7, line 150) - I find the outline of the linguistic comparisons and potential links to non-verbal cognition quite lengthy, unfocused, and confusing. I’d encourage the authors to rewrite this entire section to first outline differences and then explain how these may lead to different cognitive benefits.

Paragraph starting page 7, line 160 - this could be the end of the previous paragraph

Paragraph starting page 7, line 163 - what is the interpretation of these differences (if available)

Paragraph starting page 8, line 174 - I suggest moving this paragraph to the end of this section (essentially below the next one) and expanding on it a little - it is not entirely clear what this adds here above and beyond what has already been discussed. I also believe it should be acknowledged at this point that findings in this area are mixed.

Page 10, line 239 - I find the phrasing here rather weak given that literature delivers no clear direction. I’d argue that rather than it being ‘interesting to see what happens’, the study rather aims to provide more information as to whether alerting is affected by immersion or not.

Page 11, line 250 - Again, I don’t really think the phrasing here is ideal in terms of providing clear aims. This section also raises the question why these particular year groups were chosen. Again, I’d encourage the authors to provide clearer aims/hypotheses and to avoid the rather ambiguous notion of something being interesting to look at.

Page 11, line 264 - As noted before, the notion of something being ‘interesting’ doesn’t sit well with me - are the authors trying to establish at which point during development such benefits may occur? If so, I believe this should be clearly stated.

Page 12, line 269 - I would remove the first sentence, the second says something very similar but acknowledges that alternative findings would be possible

Response: Thank you for all your suggestions. We proceeded to the changes in reorganising and rewriting the introduction insisting first on the differences between the languages and thereafter discussing the potential cognitive consequences of these differences. We also introduced the inconsistent outcomes found in the field. We also reformulated the aims of the study especially in regard with the modifications concerning the arithmetic performances. 

Method

Page 13, line 298 - socioeconomic?

Page 14, Table 2 - where ES=0.00, should it not be ES<0.01?

Response : We proceeded to the changes, thank you. 

Results

General - I am wondering why median and not mean reaction time was analysed here, is this common practice for this task

Page 18, line 419 - if at all, I would call this marginally non-significant, especially given the Bayesian statistics - in general, it appears that the data cannot provide consistent support for alerting differences in either case

Paragraph starting page 19 , line 426 - ‘=‘ signs missing for partial eta squared and once again I find < 0.01 more meaningful than = 0.00

Page 19, line 445 - I recommend exchanging the word ‘superiority’ with a slightly more balanced/precise one, such as ‘performance advantage’ or even ‘superior WM performance’

Page 21, line 477 - it is questionable whether the comparison between English- and non-immerse children should be considered significant at p = .03 or would warrant correction which means it would not be, in either case I suggest the authors include a note that this should be interpreted with caution

Page 22, line 485 - where are these comparisons reported?

Correlations - I am not convinced that these analyses are very meaningful as they are presented, just because one correlation is significant and another is not does not mean they are significantly different (.14 and .46 are, for example, not according to a Fisher’s r-to-z transformation). I suggest the authors consider what they are really seeking to evaluate here and to reconsider the analyses accordingly. Possibly multiple regressions with dummy coding for groups would yield clearer results.

Response: 

We agreed and proceeded to the different changes. Thank you. 

Median reaction times better reflect children's performance as they do not take into account outlier reaction times. This measure has also been used in previous studies such as Barbu et al. (2019). According to us, it is a more precise measure of the real performance of the child. 

Concerning the interpretation of alerting, we agreed and changed the formulation. 

Concerning the correlation, we agreed that this analyse is not fully appropriate in this context regarding our little sample, the tasks used to evaluate arithmetic performances and the results obtained at the cognitive level (few advantages in executive functions). The main objective was to identify if CLIL context could influence arithmetic outcomes. As the literature found mixed results, we wanted to determine if the languages at stake and time spent in immersion could also play a role on the inconsistent academic performance outcomes. We modified the entire paper in this way.

Discussion

Page 25, paragraph 1 - I recommend rewriting and shortening this paragraph as there is quite a bit of repetition in it and it is not very easy to follow

We shortened the paragraph, we hope it will be clearer.

Page 26, paragraph 1 - it could be more clear why the authors consider cognitive demand to be a potential confound

Response : we did not find the idea mentioned on page 26. We hope that with the modifications made, the whole will be clearer.

Page 27 - this could be shorter - in general I feel like there is quite a bit of repetition in the discussion, the section could be more concise overall

Response : concerning these two last remarks, we tried to make the discussion more concise and more clear by resuming the general ideas already mentioned in the introduction. 

I also recommend updates in terms of a cautious evaluation of the data as appropriate and as recommended in the commentary on the results section

6. PLOS authors have the option to publish the peer review history of their article (what does this mean?). If published, this will include your full peer review and any attached files.

Reviewer #1: Yes: Ken Paap

Reviewer #2: Yes: Dr Julia Ouzia

---

## [Decision Letter · Decision Letter 1]

12 Aug 2021

PONE-D-21-09083R1

Early bilingual immersion school program and cognitive development in French-speaking children: Effect of the second language learned (English vs. Dutch) and exposition duration (2 vs. 5 years)

PLOS ONE

Dear Dr. Gillet,

I have now received feedback from both reviewers. As you can see, they both acknowledged your effort in addressing their comments in your resubmission.

Their feedback is positive, and although I fully share the view of Reviewer 1, I am willing to consider your work for publication in PLOS ONE.

However, I kindly ask you to take a look at the minor amendments that both reviewers suggest and incorporate them in your next resubmission.

Thank you very much.

Roberto Filippi

We look forward to receiving your revised manuscript.

Kind regards,

Roberto Filippi

Academic Editor

PLOS ONE

Journal Requirements:

Reviewers' comments:

Reviewer's Responses to Questions

**Comments to the Author**

1. If the authors have adequately addressed your comments raised in a previous round of review and you feel that this manuscript is now acceptable for publication, you may indicate that here to bypass the “Comments to the Author” section, enter your conflict of interest statement in the “Confidential to Editor” section, and submit your "Accept" recommendation.

Reviewer #1: (No Response)

Reviewer #2: All comments have been addressed

2. Is the manuscript technically sound, and do the data support the conclusions?

Reviewer #1: Yes

Reviewer #2: Yes

3. Has the statistical analysis been performed appropriately and rigorously? 

Reviewer #1: Yes

Reviewer #2: Yes

4. Have the authors made all data underlying the findings in their manuscript fully available?

Reviewer #1: Yes

Reviewer #2: Yes

5. Is the manuscript presented in an intelligible fashion and written in standard English?

Reviewer #1: Yes

Reviewer #2: Yes

6. Review Comments to the Author

Reviewer #1: Summary. This study tested for differences in three components of executive functioning (alerting, cognitive flexibility, and working memory) and arithmetic abilities in children (either 2nd or 5th grade) in three language groups: English immersed, Dutch immersed, and non- immersed French monolinguals. No differences between the three language groups were obtained in second grade, but in fifth grade the two immersed groups outperformed the monolingual controls on the cognitive flexibility task (but did not differ from each other). Only the Dutch immersed group outperformed the monolingual control in the working memory task. With respect to arithmetic, the Dutch immersed group outperformed the monolingual controls who, in turn, were better than the English immersed group.

In this revision the authors have been very responsive to the points raised by the reviewers and they deserve considerable credit for having done so. I would have no objection to the publication of this study as it matches the standards in the relevant literature.

However, I would say that more-of-the-same will not move the needle in resolving the many inconsistencies in this literature. The sample sizes are far too small, especially when participants cannot be randomly assigned to the conditions of interest. Furthermore, the children in the different language groups are taught by different teachers and have different peer cohorts. (Although the groups are matched on age, SES, and IQ and that’s great!) Some of the key results may be due to extraneous factors – consider the finding that only the Dutch immersed group outperformed the monolingual group in the working memory task. The authors spend a fair amount of time discussing why language similarity may sometimes lead to differences between the two immersion groups, but intuitively the differences in similarity are more subtle (see Paap, Darrow, Dalibar, & Johnson, 2015 for details) than the contrast between bilinguals and monolinguals. Yet the working memory results show no differences between French-English bilinguals and monolinguals.

The incoherent pattern of results gets worse when the present results are integrated with earlier results: “It seems that the moment of appearance of the cognitive advantages, as well as the specific cognitive function(s) enhanced, vary in function of the language learned. Moreover, once the advantage appears, it may not be sustainable. In English immersed children, advantage in selective auditory attention for example appears in first grade [3], not in second grade, reappears in third grade [2] and is not found later in CLIL schooling. The present study showed an advantage in cognitive flexibility only later in the schooling (fifth grade) confirming the outcomes of a previous study in Dutch CLIL context, that found no advantages in first, second, or third grades while advantages were found in sixth grade in cognitive flexibility and working memory[5].” This now you see it, now you don’t pattern is consistent with chronically underpowered studies. I simply have no confidence that an exact replication with sample sizes of 30 would produce the same pattern of interaction across tasks and language groups. Small sample sizes do not simply make it less likely to detect small real effects, they also make false positives more likely. I applaud Brysbaert’s 2020 recent plea (Bilingualism: Language & Cognition) for bilingual researchers to step up our game and recruit adequate sample sizes even when it is difficult and costly to do so. I suspect that closure will not come until we invest in a large N longitudinal study.

Minor comments:

p. 10 “The experimenters had to test the same number of immersed and non-immersed children.” How could this be if there are twice as many immersed participants as non-immersed?

p. 21. “anecdotic” should be anecdotal

p. 26. “Note however that some researchers suggested that there was a clear publication bias toward studies showing a bilingual advantage and suggested that this advantage do [does] not exist or is really small and/or task-dependent [51-53].” At the risk of tooting our own horn this would be a good additional study to cite here: Paap, Mason, Zimiga, Ayala-Silva, & Frost (2021). The alchemy of confirmation bias transmutes expectations into bilingual advantages: A tale of to new meta-analyses. QJEP, 73(8), 1290-1299.

Reviewer #2: I believe the manuscript is in a much better shape than in its original form, three very minor points:

p 16 line 347 - space missing('3 as' not '3as')

p 22 line 495/496 - I still somewhat stumble over the argument here - what would an alerting task requiring 'more profound cognitive processing' be?

p 26 line 580 - the shift in topic seems rather abrupt, I wonder whether a subheading would be good here

7. PLOS authors have the option to publish the peer review history of their article (what does this mean?). If published, this will include your full peer review and any attached files.

Reviewer #1: **Yes: **Ken Paap

Reviewer #2: **Yes: **Dr Julia Ouzia

---

## [Author Response · Author response to Decision Letter 1]

26 Sep 2021

We thank the reviewers and the editor for their constructive comments which allowed us to refine our manuscript.

6. Review Comments to the Author

Reviewer #1: 

Summary. This study tested for differences in three components of executive functioning (alerting, cognitive flexibility, and working memory) and arithmetic abilities in children (either 2nd or 5th grade) in three language groups: English immersed, Dutch immersed, and non- immersed French monolinguals. No differences between the three language groups were obtained in second grade, but in fifth grade the two immersed groups outperformed the monolingual controls on the cognitive flexibility task (but did not differ from each other). Only the Dutch immersed group outperformed the monolingual control in the working memory task. With respect to arithmetic, the Dutch immersed group outperformed the monolingual controls who, in turn, were better than the English immersed group.

In this revision the authors have been very responsive to the points raised by the reviewers and they deserve considerable credit for having done so. I would have no objection to the publication of this study as it matches the standards in the relevant literature.

However, I would say that more-of-the-same will not move the needle in resolving the many inconsistencies in this literature. The sample sizes are far too small, especially when participants cannot be randomly assigned to the conditions of interest. Furthermore, the children in the different language groups are taught by different teachers and have different peer cohorts. (Although the groups are matched on age, SES, and IQ and that’s great!) 

Response: We fully agree that further studies should ideally include larger samples. We added this limitation and made reference to Brysbaert (2020), page 27 of the manuscript. 

Some of the key results may be due to extraneous factors – consider the finding that only the Dutch immersed group outperformed the monolingual group in the working memory task. The authors spend a fair amount of time discussing why language similarity may sometimes lead to differences between the two immersion groups, but intuitively the differences in similarity are more subtle (see Paap, Darrow, Dalibar, & Johnson, 2015 for details) than the contrast between bilinguals and monolinguals. Yet the working memory results show no differences between French-English bilinguals and monolinguals.

Response: Concerning the advantage (or “boost”) observed for the working memory task in Dutch immersed children as compared to non-immersed and English immersed children, our main hypothesis is that the learning of languages characterized by different canonical syntactic structures – SVO and SOV – should may stimulate the ability to maintain different parts of a sequence of stimuli in a working memory task (see Amici et al., 2019): SVO structure languages would enhance the focus on final items and SOV structure language would enhance the focus on initial items. Hence, the combined learning of two languages with different structures could consequently lead to better overall recall performance in working memory tasks (better recall of first and last items of the sequence). This hypothesis needs however to be tested more directly. Another hypothesis is that this advantage could be related to the manipulation of two different number systems for the French-speaking children immersed in Dutch: a system with a unit-ten structure in Dutch and a system with a ten-unit structure in French. Particularly the Dutch structure will be challenging for French-speaking children as the linguistic order will be the opposite of the written numbers (e.g.: 21 = eenentwentig), and hence the number words need to be maintained in working memory the time they are fully processed and associated with the correct numbers while in French a more direct linear correspondence will operate (e.g., 21 = vingt-et-un)

The incoherent pattern of results gets worse when the present results are integrated with earlier results: “It seems that the moment of appearance of the cognitive advantages, as well as the specific cognitive function(s) enhanced, vary in function of the language learned. Moreover, once the advantage appears, it may not be sustainable. In English immersed children, advantage in selective auditory attention for example appears in first grade [3], not in second grade, reappears in third grade [2] and is not found later in CLIL schooling. The present study showed an advantage in cognitive flexibility only later in the schooling (fifth grade) confirming the outcomes of a previous study in Dutch CLIL context, that found no advantages in first, second, or third grades while advantages were found in sixth grade in cognitive flexibility and working memory[5].” This now you see it, now you don’t pattern is consistent with chronically underpowered studies. I simply have no confidence that an exact replication with sample sizes of 30 would produce the same pattern of interaction across tasks and language groups. Small sample sizes do not simply make it less likely to detect small real effects, they also make false positives more likely. I applaud Brysbaert’s 2020 recent plea (Bilingualism: Language & Cognition) for bilingual researchers to step up our game and recruit adequate sample sizes even when it is difficult and costly to do so. I suspect that closure will not come until we invest in a large N longitudinal study.

Response: Indeed, the effect size of the interaction is low as well as associated power. To try to qualify our point further, we added the statement 'if a cognitive advantage is demonstrated', line 482 to insist on the fact that it may not necessarily appear in other replication studies. However, we should note that since 2012, we conducted 7 studies in our lab with the same tasks (Kitap and Tab batteries) as used here and the same type of population (French speaking children immersed in Dutch or English Elementary schools), with sample sizes varying between 30 and 60 children per group. Of these studies, five found an advantage in one or more cognitive function(s) while two found no advantage. Thus, if the group differences observed in these studies would be statistical artefacts with the true effect being a no effect, then we would have expected a majority of studies showing no effect, and a few studies showing either an advantage or a disadvantage. However, the latter was never observed and the majority of studies showed an effect (although not always on the same measures). Hence, we think that a cognitive advantage could emerge over the course and context of CLIL schooling. Like mentioned above, we also added a reference of Brysbaert (2020), acknowledging that future studies should use larger sample sizes. 

Minor comments:

p. 10 “The experimenters had to test the same number of immersed and non-immersed children.” How could this be if there are twice as many immersed participants as non-immersed?

Response: We corrected the sentence; it is the same number of children in each language group. Thank you. 

p. 21. “anecdotic” should be anecdotal

Response: We changed the word, thank you! 

p. 26. “Note however that some researchers suggested that there was a clear publication bias toward studies showing a bilingual advantage and suggested that this advantage do [does] not exist or is really small and/or task-dependent [51-53].” At the risk of tooting our own horn this would be a good additional study to cite here: Paap, Mason, Zimiga, Ayala-Silva, & Frost (2021). The alchemy of confirmation bias transmutes expectations into bilingual advantages: A tale of to new meta-analyses. QJEP, 73(8), 1290-1299.

Response: We added the study of Paap et al. (2021) as well as relevant references, 

Reviewer #2: 

I believe the manuscript is in a much better shape than in its original form, three very minor points:

p 16 line 347 - space missing('3 as' not '3as')

Response: This error has been corrected, thank you! 

p 22 line 495/496 - I still somewhat stumble over the argument here - what would an alerting task requiring 'more profound cognitive processing' be?

Response: Future studies may use slightly more demanding tasks than the detection of simple visual stimuli such as an odd/even number judgement for example. The use of these more complex reaction time tasks would allow for a better characterization of the specificity the reaction time differences in tasks involving cognitive flexibility. Consequently, we added the example of an odd/even judgement task on page 23 that allows matching the groups on cognitive processing and not just on perceptive reaction time

p 26 line 580 - the shift in topic seems rather abrupt, I wonder whether a subheading would be good here

Response: We added the following subheading ‘arguments against a cognitive advantage of bilingualism’. Thank you for this suggestion. 

Other comments from the authors: 

1. All references have been checked for completeness and accuracy. Two references have been added and the reference De Wilde et al., 2017, was removed.

---

## [Editor Report · Decision Letter 2]

29 Sep 2021

Early bilingual immersion school program and cognitive development in French-speaking children: Effect of the second language learned (English vs. Dutch) and exposition duration (2 vs. 5 years)

PONE-D-21-09083R2

Dear Dr. Gillet,

We’re pleased to inform you that your manuscript has been judged scientifically suitable for publication and will be formally accepted for publication once it meets all outstanding technical requirements.

Kind regards,

Roberto Filippi

Academic Editor

PLOS ONE

---

## [Editor Report · Acceptance letter]

7 Oct 2021

PONE-D-21-09083R2 

Early bilingual immersion school program and cognitive development in French-speaking children: Effect of the second language learned (English vs. Dutch) and exposition duration (2 vs. 5 years) 

Dear Dr. Gillet:

I'm pleased to inform you that your manuscript has been deemed suitable for publication in PLOS ONE. Congratulations! Your manuscript is now with our production department. 

Kind regards, 

on behalf of

Dr. Roberto Filippi 

Academic Editor

PLOS ONE